# Optimization Model of Regional Traffic Signs for Inducement at Road Works

**Lianzhen Wang, Han Zhang, Lingyun Shi, Qingling He and Huizhi Xu \***

School of Traffic and Transportation, Northeast Forestry University, Harbin 150040, China;
wanglianzhen85@nefu.edu.cn (L.W.); wbzh@nefu.edu.cn (H.Z.); sling@nefu.edu.cn (L.S.);
qinglinghe@nefu.edu.cn (Q.H.)
**\*** Correspondence: xuhuizhiedu@nefu.edu.cn

**Abstract:** A variety of pipelines are distributed under urban roads. The upgrading of pipelines is bound to occupy certain road resources, compress the driving space of motor vehicles for a long time, aggravate the traffic congestion in the construction section, and then affect the traffic operation of the whole region. A reasonable layout of traffic signs for inducement to guide the traffic flow in the area where the construction section is located is conducive to promoting a balanced distribution of traffic flow in the regional road network, so as to achieve the reduction of automobile exhaust emissions and the sustainable development of traffic. In this paper, the layout optimization method of regional traffic signs for inducement is proposed. Taking the maximum amount of guidance information that the regional traffic signs can provide as the objective function, and taking the traffic volume, the characteristics of intersection nodes and the standard deviation of road saturation as the independent variables, the layout optimization model of guidance facilities is constructed, which can optimize the layout of traffic guidance signs in the area affected by the construction section, and achieve the goal that the minimum number of facilities can provide the maximum amount of guidance information. The results of the case study show that among the 64 alternative locations where traffic guidance signs can be set in the study area, eight optimal locations are finally determined as the setting points of guidance facilities through this model, and the effective increment of guidance information is the largest at this time. The model proposed in this paper can be used for reference to promote the sustainable development of traffic in the area where the construction section is located.

**Keywords:** traffic signs; inducement; optimization; road works; sustainable development

## 1. Introduction

In recent years, with the acceleration of China's urbanization process, urban internal construction projects are increasing, such as subway construction, pipeline construction, etc. In the process of project construction, urban road resources are often occupied in a certain period of time, which further reduces the traffic space of motor vehicles, making the urban road traffic congestion and safety problem more serious during the construction period [1]. In the construction process, the reasonable release of traffic guidance information is conducive to the balanced distribution of traffic flow on the road network, which will alleviate urban traffic congestion, reduce urban traffic pollution [2], and play an important role in promoting the sustainable development of traffic. However, at present, there is a lack of systematic research on the layout of traffic guidance information at the road works zone, which may lead to some deviation of road users' understanding of the information released by existing traffic guidance facilities, and then reduce the effectiveness of traffic guidance information during construction. Therefore, according to the traffic organization design scheme during the construction period, reasonably setting the form, location and quantity of traffic guidance facilities is conducive to improving the acceptance of guidance information and traffic operation efficiency, thus effectively alleviating traffic congestion and promoting the sustainable development of traffic.

Some analyses and research have been carried out on the optimal setup of guidance facilities in construction areas. As accurate traffic flow data is crucial for traffic control and management [3], there are considerable studies on modeling the speed–flow relationship for freeway bottlenecks. Maze et al. [4] proposed a speed–flow model in a work zone located on Interstate Highway 80 between US 61 and Interstate Highway 74. Racha et al. [5] developed a non-linear mathematical model to fit the speed–flow curve at a South Carolina work zone. Malviya et al. [6] put forward a two-stage approximation algorithm to improve the release efficiency of traffic-induced information by using predictive calculation. Senge et al. [7] proposed a path-guidance system based on swarm tracking. Weng et al. [8] investigated the speed–flow relationship and drivers' merging behavior in work zone merging areas. The results could provide accurate information for traffic engineers to calculate the merge lane length. Olia et al. [9] utilized a microscopic simulation model to demonstrate traffic flow in a network and emulate the connected vehicle systems. The result shows that connected vehicle systems can significantly assist to improve mobility and reduce traffic congestions associated with work zones. Based on VMS, Shi et al. [10] studied the dynamic regional partition problem and the dynamic traffic guidance problem under real-time traffic conditions using the traffic guidance model and put forward the architecture and two algorithms. Paz et al. [11] investigated a control method combining the rolling time domain strategy and traveler-induced information.

Zhong et al. [12] found that the road condition information shown on VMS could significantly impact the drivers' guidance compliance behaviors. Yang et al. [13] showed that the acceptance degree of guidance information is one of the key factors for driver's route choice. Alena et al. [14] investigated the effects of route guidance variable message signs (VMS) on speed and route choice in a field study on two sites on motorways. Li et al. [15] indicated that providing travel time information via VMS to travelers may degrade the network performance under some poor designs. Dia et al. [16] found that due to different individual characteristics of traffic participants, different attention is paid to traffic-induced information. Chatterjee et al. [17] found that only 61% of the individuals noticed the logo information, 92% of them read and understood the information content of the logo, but only 30% of them chose to accept the suggestion of the logo information. However, Kattan et al. [18] found that 21.4% of the individuals would not take the information suggested by the sign as the travel route, and 16.4% of the individuals would occasionally follow the advice. Chorust et al. [19] believe that if the current travel path is poor, traffic participants are more willing to accept the travel path provided by traffic-induced information. Teamar et al. [20] studied the design of guidance signs on the basis of ergonomic principles. Kirmizioglu [21] found that traffic signs can provide critical information to support safe driving in a short time, but the success rate depends on the driver's ability to understand. Changing the used and accepted signs may lead to serious safety problems. Francene [22] studied the communication efficiency of variable information signs. Erke et al. [14] studied the influence of variable information signs (VMSs) on traffic flow on closed sections of expressway, and the research results showed that VMSs are effective for traffic flow rerunning. Guo et al. [23] proposed a new optimization layout method for variable message signs based on maximization of the actual guidance effect.

Huang [24], starting from the time and place of the traffic-induced information setting, constructed a model to determine the area affected by the release of induced information. Wang et al. [25] found that the induced information could influence the driver's behavior and analyzed the data. After studying the process of drivers' recognition, Zhou [26] drew up a plan for the design and arrangement of signs in the expressway construction area. Wang [27] pointed out in combination with the setting regulations of traffic signs and markers that existing signs should avoid information overload and should be adjusted continuously according to the actual situation. The location should also be set in an appropriate place that can attract the attention of drivers. Zheng [28] analyzed the standardization of the information release form of traffic guidance facilities and put forward the standard model. Lv et al. [29] further studied how to design guidance facilities

and traffic organization schemes in construction roads. Yi et al. [30] took into account that the variable information board and static traffic sign have different recognition times under different installation environments and made improvements. Li [31] studied the information overload threshold of facilities in combination with the traffic characteristics of specific areas.

Domestic and foreign research on the setting of guidance facilities mainly focus on the influence of induced information on behavior, content design and release forms, using many available methods [32], while the research on its setting location is less extensive. In this paper, the existing model is improved to increase the weight of section traffic volume, node importance and standard deviation of section saturation that can choose the driving direction, and the optimization model of setting the location of traffic guidance facilities is built to determine the optimal setting point position.

## 2. Data Collection

According to the factors affecting the installation of traffic guidance facilities and the traffic characteristics of the road construction area, the study selected the node importance, traffic volume and road saturation standard deviation as the independent variables of the model.

### 2.1. Node Characteristics

The road network nodes in the construction area are shown in Figure 1, wherein the real line section is the construction section—that is, the path composed of nodes 12, 13, 14, 15, 17, 18, 4, 3, 2, 27 and 29. The rest are the unblocked roads, and the arrow direction represents the driving direction. The node features of 29 nodes in the road network in the construction area were counted according to three indicators—namely, whether they were decision points $x_1$, the number of decision points excluding the construction section $x_2$, and the node $x_3$ intersecting with the construction road. After removing the irrelevant nodes, 23 nodes were finally determined, and the statistical results are shown in Table 1.

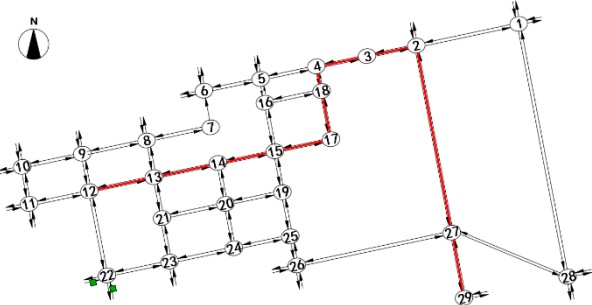

**Figure 1.** Node locations of road network in construction area.

**Table 1.** Nodes feature statistics.

| ID | Decision Point or Not (1—Yes, 0—No) $x_1$ | The Number of Decision Points Excluding the Construction Section $x_2$ | The Number of Nodes That Intersect with the Construction Road $x_3$ |
|---|---|---|---|
| 1 | 1 | 3 | 1 |
| 2 | 1 | 2 | 0 |
| 3 | 0 | 0 | 0 |
| 4 | 1 | 1 | 0 |
| 5 | 1 | 3 | 1 |
| 8 | 1 | 3 | 1 |
| 9 | 1 | 3 | 1 |
| 11 | 1 | 3 | 1 |
| 12 | 1 | 3 | 0 |
| 13 | 1 | 2 | 0 |

**Table 1.** *Cont.*

| ID | Decision Point or Not (1—Yes, 0—No) $x_1$ | The Number of Decision Points Excluding the Construction Section $x_2$ | The Number of Nodes That Intersect with the Construction Road $x_3$ |
|---|---|---|---|
| 14 | 1 | 1 | 0 |
| 15 | 1 | 2 | 0 |
| 16 | 1 | 3 | 1 |
| 17 | 0 | 0 | 0 |
| 18 | 1 | 1 | 0 |
| 19 | 1 | 2 | 1 |
| 20 | 1 | 3 | 1 |
| 21 | 1 | 2 | 1 |
| 22 | 1 | 3 | 1 |
| 26 | 1 | 3 | 1 |
| 27 | 1 | 2 | 0 |
| 28 | 1 | 3 | 1 |
| 29 | 0 | 0 | 0 |

*2.2. Traffic Volume*

The traffic volume data of the construction road and the unblocked roads at the evening peak period (17:00 to 19:00) before and during the construction period were investigated and counted, obtaining the traffic volume data of 29 intersections and 40 road sections in the construction affected area. The investigation method was a manual investigation. The continuous surveys were conducted from Monday to Friday.

*2.3. Section Saturation*

The ratio of the actual traffic flow and the actual capacity of the section is used to calculate the saturation of each section in the construction area. The actual traffic capacity is calculated based on the basic capacity and combined with the actual situation of the construction road, different correction factors are considered in the calculation process, including the reduction factors of lane number, lane width and intersection distance.

**3. Model Construction and Solution**

*3.1. Existing Models*

Han et al. [33] built a decision point optimization model for the layout of passenger guide signs in large passenger stations with the goal of maximizing the information amount of regional guidance. Xu et al. [34] modified the existing model and increased the distance between two adjacent information query desks within the visible range of passengers as a constraint condition. At the same time, considering the weight of passenger flow and the importance of nodes, they built an optimal information query layout model for subway stations and solved the optimal information query layout position; the model is as follows [34]:

$$maxZ = \sum_{j=1}^{m} \sum_{i_1=1}^{n} \frac{f(j)a_{j,i_1}(\omega'_{j,i_1})^\alpha}{N_{j,i_1}} w_l + \sum_{j=1}^{m} \sum_{i_2=1}^{q} \frac{f(j)a_{j,i_2}(\omega'_{j,i_2})^\alpha}{N_{j,i_1}} w_l + \sum_{j=1}^{m} c_j w_p \qquad (1)$$

The number of points set by the guide sign is:

$$\sum_{j=1}^{m} f(j) = S, j = 1, 2, \text{L}, m \qquad (2)$$

where: $i$ is the $i$-th target point, $i_1$ is the number of inbound target points, and $i_2$ is the number of outbound target points; $j$ is the $j$-th identification alternative point; $m$ is the number of alternative points of signs; $n$ is the number of target points—that is, the number of destinations that passengers want to reach; $f(j)$ is a variable of 0 to 1. When the

information inquiry station is set at the $j$-th identification alternative point, the value is 1; otherwise, it is 0. $a_{j,i}$ is the distribution coefficient; when the $i$-th target point is guided by the $j$-th candidate point, the value is 1; otherwise, it is 0. $\omega'_{j,i}$ is the normalized value of passenger flow from the $j$-th alternative point to the $i$-th target point, $\omega'_{j,i1}$ is the normalized value of inbound passenger flow, $\omega'_{j,i2}$ is the normalized value of outbound passenger flow; $\alpha$ is the weight relationship between the induced passenger flow and the induced distance of the information inquiry station, taking 1.5; $N_{j,i}$ is the number of crossing points from the $j$-th identification candidate point to the $i$-th target point; $\omega_1$ is the weight of passenger flow; $\omega_p$ is the node weight; $c_j$ is the importance of node $j$; $S$ is the number of planning signs.

The above model well reflects the relationship between the location of the information inquiry desk in the subway station and the guidance passenger flow at the target point, the number of crossing points and the importance of nodes and provides good theoretical support for determining the location of the information inquiry desk in the subway station and then reasonably guiding the passenger flow. This paper intends to draw lessons from the advantages of this model in determining the maximum information of the guidance facilities and study the setting of the guidance facilities in the regional road network during urban road construction, so as to determine the optimal setting location and quantity of the guidance facilities.

### 3.2. Model Building

The largest amount of induced information should be considered in the location of guidance facilities in the construction area, and the induced information is related to the road traffic volume, node importance and the standard deviation of road saturation in the optional driving direction.

Considering the standard deviation of road saturation with the optional driving direction, the location layout model of guidance facilities with the goal of maximizing the induced information was established.

$$\max Z = \sum_{j=1}^{m} f(j)[\omega'_j w_1 + c_j w_p + \sigma'_j w_\sigma] \tag{3}$$

The number of points set by the guidance facility is:

$$\sum_{j=1}^{m} f(j) = S, j = 1, 2, \ldots, m \tag{4}$$

where: $j$: alternative locations for induction facility settings; $m$: the total number of alternative points; $f(j)$: 0 to 1 variable—when the induction facility is set at the alternative position $j$, the value is 1; otherwise, it is 0. $\omega'_j$: the normalized value of the traffic induced by the guidance facility at the $j$ alternative location; $c_j$: node importance; $\sigma'_j$: the normalized value of the road saturation standard deviation of the alternative driving direction of the alternative point; $w_1$: the weight of section traffic; $w_p$: the weight of the node; $w_s$: the weight of the standard deviation of section saturation; $S$: the total number of traffic induction facilities to be set up.

In Formula (3), the objective function is related to the normalized value of guidance traffic volume, node importance, the normalized value of the standard deviation of road saturation of alternative points with the driving direction, and the respective weights of the three. The larger the normalized processing value of guidance traffic volume, the more drivers receive the information and the higher the efficiency of the guidance facilities. The greater the importance of nodes, the stronger the importance of alternative points in setting guidance facilities in the regional road network. The greater the normalized value of road saturation standard deviation of alternative points that can choose the driving direction, the greater the difference of road saturation ahead. Considering factors such as time efficiency and driver's mental state, guidance facilities should be set up here to signal the condition of

the driver's road ahead and to consider factors such as the time cost by drivers themselves deciding to choose another way, to establish a greater role of guidance facilities. Therefore, when the sum of the normalized value of traffic volume, node importance and the standard deviation of road saturation of alternative points that can choose the driving direction is the maximum, the effect of guidance facilities can be maximized. The constraint condition Formula (4) represents the constraint of the total number of locations of guidance facilities.

*3.3. Model Solving*

(1) Calculate the importance of the node

Step 1: consistency treatment.

$$x'_i = 1 - x_i \qquad (5)$$

where $x'_i$ is the value of $x_i$ after consistency treatment. $x_1, x_2$ are economic indices, and $x_3$ is the cost index, which needs consistency treatment. The results of consistency treatment are shown in Table 2.

**Table 2.** The results of consistency treatment.

| ID | Decision Point or Not (1—Yes, 0—No) $x_1$ | The Number of Decision Points Excluding the Construction Section $x_2$ | The Number of Nodes That Intersect with the Construction Road $x_3$ |
|---|---|---|---|
| 1 | 1 | 3 | 0 |
| 2 | 1 | 2 | 1 |
| 3 | 0 | 0 | 1 |
| 4 | 1 | 1 | 1 |
| 5 | 1 | 3 | 0 |
| 8 | 1 | 3 | 0 |
| 9 | 1 | 3 | 0 |
| 11 | 1 | 3 | 0 |
| 12 | 1 | 3 | 1 |
| 13 | 1 | 2 | 1 |
| 14 | 1 | 1 | 1 |
| 15 | 1 | 2 | 1 |
| 16 | 1 | 3 | 0 |
| 17 | 0 | 0 | 1 |
| 18 | 1 | 1 | 1 |
| 19 | 1 | 2 | 0 |
| 20 | 1 | 3 | 0 |
| 21 | 1 | 2 | 0 |
| 22 | 1 | 3 | 0 |
| 26 | 1 | 3 | 0 |
| 27 | 1 | 2 | 1 |
| 28 | 1 | 3 | 0 |
| 29 | 0 | 0 | 1 |

Step 2: dimensionless treatment, obtained H = ATA

$$X' = \frac{x - \bar{x}}{\sigma} \qquad (6)$$

where $x$ is the evaluation value, $\bar{x}$ is the average value, and $\sigma$ is the standard deviation. The results of dimensionless treatment are shown in Table 3.

**Table 3.** The results of dimensionless treatment.

| ID | Decision Point or Not (1—Yes, 0—No) $x_1$ | The Number of Decision Points Excluding the Construction Section $x_2$ | The Number of Nodes that Intersect with the Construction Road $x_3$ |
|---|---|---|---|
| 1 | 0.39 | 0.86 | 0.96 |
| 2 | 0.39 | −0.08 | −1.04 |
| 3 | −2.58 | −1.97 | −1.04 |
| 4 | 0.39 | −1.03 | −1.04 |
| 5 | 0.39 | 0.86 | 0.96 |
| 8 | 0.39 | 0.86 | 0.96 |
| 9 | 0.39 | 0.86 | 0.96 |
| 11 | 0.39 | 0.86 | 0.96 |
| 12 | 0.39 | 0.86 | −1.04 |
| 13 | 0.39 | −0.08 | −1.04 |
| 14 | 0.39 | −1.03 | −1.04 |
| 15 | 0.39 | −0.08 | −1.04 |
| 16 | 0.39 | 0.86 | 0.96 |
| 17 | −2.58 | −1.97 | −1.04 |
| 18 | 0.39 | −1.03 | −1.04 |
| 19 | 0.39 | −0.08 | 0.96 |
| 20 | 0.39 | 0.86 | 0.96 |
| 21 | 0.39 | −0.08 | 0.96 |
| 22 | 0.39 | 0.86 | 0.96 |
| 26 | 0.39 | 0.86 | 0.96 |
| 27 | 0.39 | −0.08 | −1.04 |
| 28 | 0.39 | 0.86 | 0.96 |
| 29 | −2.58 | −1.97 | −1.04 |

The standardized matrix could be obtained: $H = A^{T}A$; the results are as follows:

$$H = \begin{bmatrix} 23.01 & 17.54 & -7.74 \\ 17.54 & 23.00 & -13.46 \\ -7.74 & -13.46 & 23.28 \end{bmatrix}$$

Step 3: programming problem solving.

$$\max c_i^{T} H c_i \tag{7}$$

When $c_i$ is positive, the eigenvector corresponding to the maximum eigenvalue of matrix $H$ is $c_i$. In this case, the maximum value of Equation (7) is obtained and then normalized. The result is as follows:

$$c_i = (0.380, 0.081, 0.539)^{T}$$

Step 4: calculate node importance.

$$c_i = \sum_{i=1}^{3} c_i x_{ij} / \max \sum_{i=1}^{3} c_i x_{ij} \tag{8}$$

where $x_{ij}$ is the evaluation of estimate. The results of node importance are shown in Table 4.

(2) Traffic volume normalization:

$$\omega_j{}' = \frac{\sum \omega_j}{\max \sum \omega_j} \tag{9}$$

(3) The weights were calculated by an analytic hierarchy process (AHP) and the consistency test was carried out.

**Table 4.** Node importance calculation result.

| ID | $x_1$ 0.380 | $x_2$ 0.081 | $x_3$ 0.539 | $c_j$ |
|---|---|---|---|---|
| 1 | 1 | 3 | 0 | 0.536 |
| 2 | 1 | 2 | 1 | 0.930 |
| 3 | 0 | 0 | 1 | 0.464 |
| 4 | 1 | 1 | 1 | 0.861 |
| 5 | 1 | 3 | 0 | 0.536 |
| 8 | 1 | 3 | 0 | 0.536 |
| 9 | 1 | 3 | 0 | 0.536 |
| 11 | 1 | 3 | 0 | 0.536 |
| 12 | 1 | 3 | 1 | 1.000 |
| 13 | 1 | 2 | 1 | 0.930 |
| 14 | 1 | 1 | 1 | 0.861 |
| 15 | 1 | 2 | 1 | 0.930 |
| 16 | 1 | 3 | 0 | 0.536 |
| 17 | 0 | 0 | 1 | 0.464 |
| 18 | 1 | 1 | 1 | 0.861 |
| 19 | 1 | 2 | 0 | 0.466 |
| 20 | 1 | 3 | 0 | 0.536 |
| 21 | 1 | 2 | 0 | 0.466 |
| 22 | 1 | 3 | 0 | 0.536 |
| 26 | 1 | 3 | 0 | 0.536 |
| 27 | 1 | 2 | 1 | 0.930 |
| 28 | 1 | 3 | 0 | 0.536 |
| 29 | 0 | 0 | 1 | 0.464 |

## 4. Model Application

### 4.1. Calculate Node Importance

According to Equations (5)–(8) and data in Table 1, the node importance of 23 nodes have been calculated, and the results are shown in Table 4. The larger the value is, the greater the importance of node $j$. The node importance of nodes 2, 4, 12, 13, 14, 15, 18, 27 are all more than 0.85, and all of these nodes are located on the construction section. The importance of the remaining nodes ranged from 0.4 to 0.6.

### 4.2. Determination of Index Weight

The weight of parameters in Equation (3) was determined by analytic hierarchy process (AHP), and the judgment matrix was determined after consulting seven experts' opinions, as shown in Table 5.

**Table 5.** AHP comparison matrix.

| | Traffic Volume | Node Importance | Standard Deviation of Road Saturation |
|---|---|---|---|
| Traffic volume | 1 | 1/1.75 | 3 |
| Node importance | 1.75 | 1 | 5 |
| Standard deviation of road saturation | 1/3 | 1/5 | 1 |

After normalization, the maximum eigenvector $W = (0.327, 0.562, 0.111)^T$ is obtained, and the maximum eigenvalue is $\lambda$max = 3.0002. Then, a consistency test is performed: $CI$ = 0.0001, the matrix $RI$ is 0.52, $CR$ = 0.00019 < 0.1, to pass the consistency test. Then, the weights of traffic volume, node importance and road saturation are $w_j$ = 32.7%, $w_p$ = 56.2% and $w_s$ = 11.1%, respectively.

### 4.3. Calculation of Parameters of Objective Function at Alternative Points

There are four, three and two entrance directions, respectively, at the intersections in the region. According to the relationship with the location of the construction road, 64 alternative points for setting guidance facilities are selected, as shown in Figure 2. The solid line represents the road works.

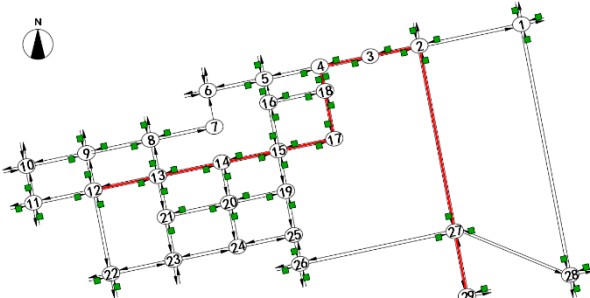

**Figure 2.** Alternative nodes for traffic guidance facilities.

According to Equation (3), the objective function values of each alternative point are analyzed and calculated, and the results are shown in Table 6.

**Table 6.** Alternative point objective function parameter calculation.

| NO | Nodes Location | Weights | | | Z |
|---|---|---|---|---|---|
| | | Traffic Volume | Node Importance | Standard Deviation of Road Saturation | |
| | | 32.7% | 56.2% | 11.1% | |
| 1 | north–1 | 0.293 | 0.536 | 0.394 | 0.441 |
| 2 | east–1 | 0.324 | 0.536 | 0.347 | 0.446 |
| 3 | 1–28 | 0.416 | 0.536 | 0.277 | 0.468 |
| 4 | 28–1 | 0.416 | 0.536 | 0.421 | 0.484 |
| 5 | north–2 | 0.403 | 0.930 | 0.379 | 0.697 |
| 6 | 1–2 | 1.000 | 0.930 | 0.419 | 0.896 |
| 7 | 2–3 | 0.556 | 0.464 | 0.000 | 0.443 |
| 8 | 3–2 | 0.541 | 0.930 | 0.412 | 0.745 |
| 9 | 2–27 | 0.616 | 0.930 | 0.673 | 0.799 |
| 10 | 27–2 | 0.483 | 0.930 | 0.068 | 0.688 |
| 11 | 3–4 | 0.633 | 0.861 | 0.554 | 0.752 |
| 12 | 4–3 | 0.619 | 0.464 | 0.000 | 0.463 |
| 13 | 5–4 | 0.629 | 0.861 | 0.610 | 0.757 |
| 14 | 4–18 | 0.052 | 0.861 | 0.294 | 0.533 |
| 15 | 18–4 | 0.025 | 0.861 | 0.056 | 0.498 |
| 16 | north–5 | 0.416 | 0.536 | 0.239 | 0.464 |
| 17 | 6–5 | 0.641 | 0.536 | 0.218 | 0.535 |
| 18 | 5–16 | 0.570 | 0.536 | 0.149 | 0.504 |
| 19 | 7–8 | 0.068 | 0.536 | 0.403 | 0.368 |
| 20 | north–8 | 0.206 | 0.536 | 0.401 | 0.413 |
| 21 | 8–9 | 0.054 | 0.536 | 0.383 | 0.361 |
| 22 | 9–8 | 0.168 | 0.536 | 0.211 | 0.380 |
| 23 | 8–13 | 0.260 | 0.930 | 0.065 | 0.615 |
| 24 | 10–9 | 0.179 | 0.536 | 0.265 | 0.389 |
| 25 | north–9 | 0.077 | 0.536 | 0.376 | 0.368 |
| 26 | 9–12 | 0.186 | 1.000 | 0.440 | 0.672 |
| 27 | 10–11 | 0.343 | 0.536 | 0.372 | 0.455 |
| 28 | west–11 | 0.198 | 0.536 | 0.936 | 0.470 |
| 29 | south–11 | 0.087 | 0.536 | 0.830 | 0.422 |
| 30 | 11–12 | 0.149 | 1.000 | 0.254 | 0.639 |

**Table 6.** *Cont.*

| NO | Nodes Location | Weights | | | Z |
| | | Traffic Volume | Node Importance | Standard Deviation of Road Saturation | |
| | | 32.7% | 56.2% | 11.1% | |
|----|----------------|----------------|-----------------|---------------------------------------|-------|
| 31 | 12–13 | 0.382 | 0.930 | 0.060 | 0.654 |
| 32 | 22–12 | 0.303 | 1.000 | 0.432 | 0.709 |
| 33 | 21–13 | 0.386 | 0.930 | 0.015 | 0.651 |
| 34 | 13–14 | 0.355 | 0.861 | 0.786 | 0.687 |
| 35 | 14–13 | 0.311 | 0.930 | 0.057 | 0.631 |
| 36 | 14–15 | 0.410 | 0.930 | 0.566 | 0.720 |
| 37 | 15–14 | 0.363 | 0.861 | 0.987 | 0.712 |
| 38 | 20–14 | 0.314 | 0.861 | 0.201 | 0.609 |
| 39 | 15–17 | 0.142 | 0.464 | 0.000 | 0.307 |
| 40 | 17–15 | 0.131 | 0.930 | 0.549 | 0.626 |
| 41 | 15–16 | 0.493 | 0.536 | 0.959 | 0.569 |
| 42 | 16–15 | 0.501 | 0.930 | 0.589 | 0.752 |
| 43 | 19–15 | 0.605 | 0.930 | 0.519 | 0.778 |
| 44 | 16–18 | 0.056 | 0.861 | 0.064 | 0.509 |
| 45 | 18–16 | 0.093 | 0.536 | 0.068 | 0.339 |
| 46 | 17–18 | 0.042 | 0.861 | 0.229 | 0.523 |
| 47 | 18–17 | 0.030 | 0.464 | 0.000 | 0.271 |
| 48 | 19–20 | 0.190 | 0.536 | 0.923 | 0.466 |
| 49 | 20–19 | 0.135 | 0.466 | 0.244 | 0.333 |
| 50 | 25–19 | 0.500 | 0.466 | 0.388 | 0.468 |
| 51 | 20–21 | 0.190 | 0.466 | 0.424 | 0.371 |
| 52 | 21–20 | 0.138 | 0.536 | 1.000 | 0.457 |
| 53 | 24–20 | 0.203 | 0.536 | 0.855 | 0.463 |
| 54 | 23–21 | 0.158 | 0.466 | 0.100 | 0.325 |
| 55 | west–22 | 0.357 | 0.536 | 0.606 | 0.485 |
| 56 | south–22 | 0.257 | 0.536 | 0.595 | 0.451 |
| 57 | west–26 | 0.592 | 0.536 | 0.204 | 0.517 |
| 58 | south–26 | 0.530 | 0.536 | 0.282 | 0.506 |
| 59 | 25–26 | 0.620 | 0.536 | 0.280 | 0.535 |
| 60 | 26–27 | 0.653 | 0.930 | 0.209 | 0.759 |
| 61 | east–28 | 0.200 | 0.536 | 0.194 | 0.388 |
| 62 | north–28 | 0.293 | 0.536 | 0.338 | 0.435 |
| 63 | 28–27 | 0.709 | 0.930 | 0.308 | 0.789 |
| 64 | east–29 | 0.351 | 0.464 | 0.000 | 0.376 |

### 4.4. Sensitivity Analysis of the Solution Results

According to Equation (4), $S$ is not the only solution. When the value of $S$ changes, the $Z$ value of induced information will change; the greater the value of $S$, the higher the $Z$ value will be. In other words, when guidance facilities are set at all alternative points, the maximum induced information will be obtained. Figure 3 shows the change of induced information of a single guidance facility under different $S$ values.

The optimal solution is different for different values of $S$; if only considering that the information-induced capacity is the largest, then $S$ takes the maximum. The setup of the guidance facility requires material cost and labor cost, so it cannot be set at each alternative point; only a part of alternative points can be selected to set the guidance sign. In order to determine the optimal value of $S$, the mean of the induced information is taken and the mean value graph and mean slope graph are drawn, as shown in Figures 4 and 5.

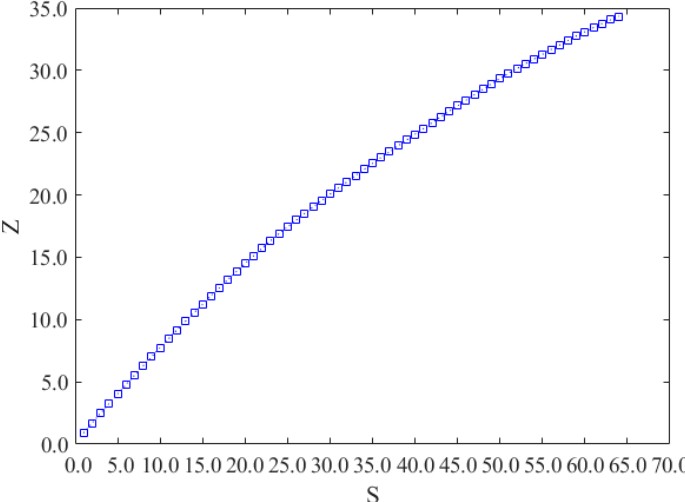

**Figure 3.** The change of information induced with *S* value. The vertical axis represents the amount of induced information Z, and the horizontal axis represents *S*.

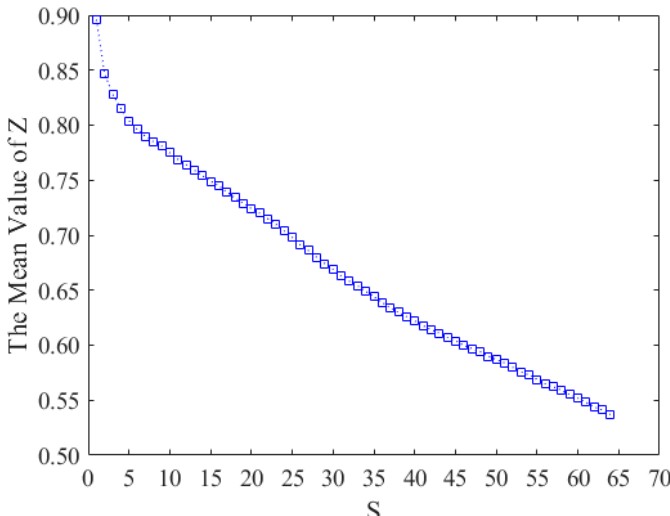

**Figure 4.** The mean value of induced information.

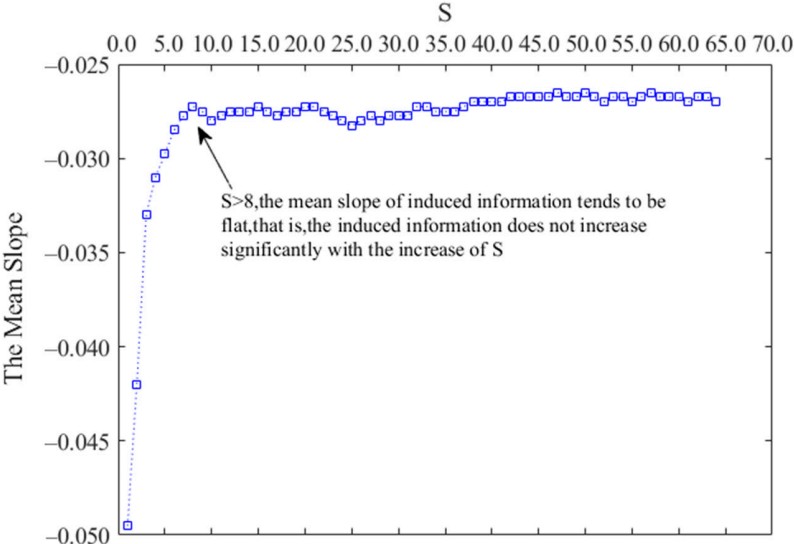

**Figure 5.** Slope diagram of mean value of induced information.

As can be seen from Figure 4, the larger the *S* value is, the smaller the induced information amount of a single guidance facility is. It can be seen from Figure 5 that when *S* > 8, the mean slope of induced information tends to be flat—that is, the induced information does not increase significantly with the increase of *S*. Therefore, when *S* = 8, the number of guidance facilities is the best, and the specific setting points are 6, 9, 63, 43, 60, 13, 11 and 42, as shown in Figure 6.

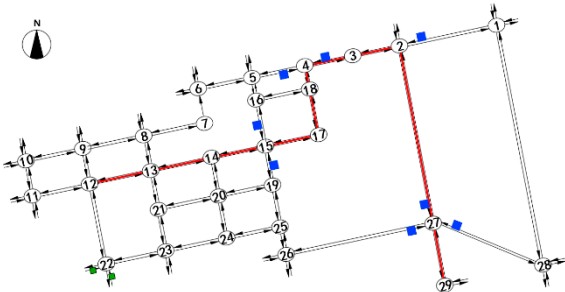

**Figure 6.** Optimal setting point of traffic guidance facilities.

## 5. Conclusions

The construction period of urban road occupation has a great impact on the regional traffic environment. Reasonable layout of traffic guidance facilities is conducive to promoting the balanced distribution of traffic flow in the regional road network. However, at present, there is a lack of systematic research on the layout of traffic guidance information at the road works zone. In this paper, the layout optimization method of regional traffic signs for inducement was proposed. The model has taken the maximum amount of guidance information that the regional traffic signs can provide as the objective function, and the traffic volume, the characteristics of intersection nodes and the standard deviation of road saturation as the independent variables, aiming to optimize the layout of traffic guidance signs in the area affected by the construction section. The paper also made a case study, the results of which showed that among the 64 alternative locations where traffic guidance signs can be set in the study area, eight optimal locations are finally determined as the setting points of guidance facilities through this model, and the effective increment of guidance information is the largest at this time. The proposed method could provide the maximum amount of guidance information for regional traffic flow using the least number of traffic signs. The research could provide reference for the location of traffic signs for inducement in the regional road network during construction and could thus reduce congestion.

**Author Contributions:** Writing—original draft preparation, L.W.; investigation, H.Z.; data curation, L.S. data curation, Q.H.; writing—review and editing, H.X. All authors have read and agreed to the published version of the manuscript.

**Funding:** This research was funded by NATIONAL NATURAL SCIENCE FOUNDATION OF CHINA, grant number 71701041, NATURAL SCIENCE FOUNDATION OF HEILONGJIANG PROVINCE, grant number LH2019E007, the Fundamental Research Funds for the Central Universities, grant number 2572019BG02, the China Postdoctoral Science Foundation, grant number 2015M581412, and the Ministry of Education of Humanities and Social Science Project, grant number 18YJAZH106.

**Institutional Review Board Statement:** Not applicable.

**Informed Consent Statement:** Not applicable.

**Conflicts of Interest:** The authors declare no conflict of interest.

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
