# Peer review of "Optimization Model of Regional Traffic Signs for Inducement at Road Works"

_sustainability, doi:10.3390/su13136996_

Round 1
Reviewer 1 Report
(i) The title is too general to clearly display the contributions of the paper. What’s more, the title is more like that of one proposal.
(ii) The language should be polished.
(iii) The citation styles of some references in the text are nonstandard and those of some references are inconsistent. For example, “Wang J.” should be written as “Wang et al.”; “Y Yang” should be written as “Yang”.
(iv) The physical meanings of many variables are not defined.
(v) The authors did introduce their proposed model, but the reviewer does not understand how they constructed this model or what contributions this model has.
(vi) The results are too simple to clearly display the contributions of this paper.
(vii) The conclusions are too simple.
(viii) The format of references is messy.
Reviewer 2 Report
The paper provides a problem statement and solution for placing a limited number of traffic signs (of many available locations). The Authors suggest a composite indicator for every location that includes traffic flows, saturation, expert-estimated weight, etc. and optimize the selection as a linear function.
In my opinion, there are two major issues of this approach:
- The composite indicator is suggested without any supporting information. This is not clear from the paper, why this indicator is better than similar indicators, introduced in existing studies, or than myriads of alternative indicators.
- The linear function assumes independent selection of the input (decision) variables, which seems not the case for placing road signs: if a driver is already informed about road works, the utility of the next signs on his routes is much smaller. The traffic flows usually have regular routes, and, without taking these dependencies into account, the solution should tend to placing all road signs on the main arterial road.
Minor issues:
- All used variables should be introduced, otherwise their meaning is unclear to the Reader (e.g., equations (1) and (2))
- It seems that model solving requires additional description – some procedures are not obvious from the first glance (e.g., consistency treatment (5), H=ATA, etc.)
- Notations are inaccurate – e.g., road saturation standard deviation is referred as sigma_j (3) or S_j (6)
- Formatting of in-text references like ”Han Y X et al.” – shouldn’t it be just “Han et al.”?
Reviewer 3 Report
This study focused on the sustainable development of road traffic. The authors tried to devise a suitable layout of traffic signs for inducement to facilitate the proper flow of traffic in under-construction areas. For this purpose, the authors proposed a layout optimization method of regional traffic signs for inducement by using the maximum amount of guidance information as an objective function. The authors claimed that the proposed method could optimize the layout of traffic guidance signs in areas affected by the construction section, which ultimately reduces the automobile exhaust emissions and provides the maximum amount of guidance information by using the minimum number of facilities. Among 64 alternative locations, the authors selected eight optimal locations and claimed that they provide maximum guidance information.
Although this paper has much information for the scientific community, similar work has already been published in previous literature, which makes the novelty of this paper doubtful. Moreover, some considerations need to be clarified. Specifically, the authors could not explain how the idea presented in this manuscript is different from those shown in previous literature.
Major Comments:
- To induce the urban traffic flow properly, Yirong et al. (2017) proposed a new optimization layout method for variable message signs based on the maximization of the actual guidance information. The proposed model by maximizing actual guidance information (Yirong et al., 2017) is similar to those presented in the current study. Overall, the main findings of the current manuscript have already been published in previous literature, which makes the novelty of this paper doubtful.
- How did you evaluate the effectiveness of traffic guide signs? Which metric have you used? There is no such information throughout the manuscript.
- In various traffic flow applications, including traffic simulation, real-time estimation, and prediction, one requires a probabilistic model of traffic flow since traffic flow is highly stochastic in nature. How the authors integrated this information into the current manuscript?
- In Line 47, the authors mentioned, “Some analyses and researches have been carried out on the optimal setup of guidance facilities in construction area.” However, they did not cite those works. It would be helpful to the scientific community if those similar works were cited and how current work is different from those published work.
- Although traffic signs provide critical information for a safe driving experience (Erkut et al., 2012), their success rate highly depends on drivers' understanding ability. Don't you think that if you optimize traffic signs and reduce them to a minimum low from over 60, it may confuse the drivers, which could cause serious safety issues?
- There are some inconsistencies in Fig. 3, 4 where it is challenging to read the x and y-axis due to the small font size. Therefore, consistency in the ranges of font sizes must be adopted to present the results more effectively. Also, the values of the x and y-axis in Fig 5 is shown with both integer and decimal points which must be changed to only decimal points in order to maintain consistency. This inconsistency in the results section raises many questions on the originality of the manuscript. I recommend to authors thoroughly re-organize the manuscript for improved readability and presentation.
Reference:
GUO Yirong, WANG Xiaoming, WANG Min, ZHANG Hong. A new method for variable message sign layout based on the actual guidance effect maximization [J]. Journal of University of Science and Technology of China, 2017, 47(4): 342-349 http://just.ustc.edu.cn/EN/Y2017/V47/I4/342
Reviewer 4 Report
Dear Authors, very interesting article. Congratulations! Below are some of my observations.
- very good literature review,
- the size of individual elements in the mathematical formulas is disproportionate to each other - I propose to open individual formulas and restore their factory settings regarding the size of symbols (sums, brackets),
- the symbols used in formulas 1 and 2 were not explained as soon as they appeared,
- I propose to expand the section on conclusions - in general; it mainly contains a summary; there are no conclusions from the conducted research.
Round 2
Reviewer 1 Report
The authors have made some modifications based on the reviewer's comments. However, there are some minor mistakes in the full text. For example, the author's information in the first references is incorrect, please check the first name and the last name and correct the errors.
Reviewer 4 Report
Thanks for making corrections.
